# High-time resolution analysis of meridional tides in the upper mesosphere and lower thermosphere at mid-latitudes measured by the Falkland Islands SuperDARN radar

Gareth Chisham[1], Andrew J. Kavanagh[1], Neil Cobbett[1], Paul Breen[1], and Tim Barnes[1]

[1]British Antarctic Survey, Cambridge, UK.

**Correspondence:** Gareth Chisham (gchi@bas.ac.uk)

**Abstract.** Solar tides play a major role in the dynamics of the upper mesosphere and lower thermosphere (MLT). Hence, a comprehensive understanding of these tides is important for successful modelling of the MLT region. Most ground-based observations of tidal variations in the MLT have been from meteor radar measurements with a temporal resolution of 1 hr. Here, we take a different perspective on these tidal variations using high-resolution 1-min neutral wind measurements from the Falkland Islands SuperDARN radar. This analysis shows that these higher-resolution data can be used to identify higher frequency tidal components than are typically observed by meteor radars (up to a heptadiurnal component). It also shows evidence of significant power in these higher frequency components, particularly in the quaddiurnal component, which may be particularly suitable for a global analysis using high-resolution SuperDARN neutral wind measurements. The high-resolution analysis also shows evidence of fluctuations with a frequency of 1.5 cycles/day, as well as higher frequency fluctuations, accompanying a quasi-two-day wave. We discuss the limitations of this high-resolution analysis method, and the new opportunities that it may provide. We conclude that higher-resolution SuperDARN neutral wind measurements need to be better exploited in the future, as they provide a complementary way of studying tides and waves in the MLT.

## 1 Introduction

In the upper mesosphere and lower thermosphere (MLT), solar tides typically comprise a significant component of the atmospheric motion. These tides are predominantly driven by lower atmospheric processes that result from solar insolation; solar radiation is absorbed by both ozone in the stratosphere and water vapour in the troposphere. These processes result in large-scale atmospheric tidal waves at integral harmonics of a solar day that propagate upward toward the MLT, increasing in amplitude as the density of the atmosphere decreases with increasing altitude (Forbes, 1995; Chapman and Lindzen, 2012). As a consequence, the tides play a major role in MLT dynamics, modulating gravity wave fluxes and redistributing energy and momentum within the MLT. Hence, observations and modelling of these tides are crucial to improve the understanding of the dynamics of the MLT, and the modelling of processes there (e.g., Griffith et al., 2021).

The motion of the MLT, and the identification of tidal variations in that motion, is a key measurement made by ground-based Very High Frequency (VHF) meteor radars (Hocking et al., 2001). Meteor radars exploit the fact that radio waves are backscattered by ionised meteor trails left by meteoroids as they ablate in the upper atmosphere (Ceplecha et al., 1998). These meteor trails drift under the influence of winds in the neutral atmosphere, and so can be used to track the motion of the MLT. Hall et al. (1997) first showed that most near-range (<400 km) echoes measured by the High Frequency (HF) radars that comprise the Super Dual Auroral Radar Network (SuperDARN) are also the result of scatter from meteor trails, enabling the use of the global network of SuperDARN radars to study winds and tides in the MLT (van Caspel et al., 2020; Hibbins et al., 2019, 2011).

Owing to the sporadic nature of the occurrence of meteors in the MLT, most studies have used spatial and temporal averages of meteor neutral wind measurements to study large-scale fluctuations in the MLT. Meteor radars sound vertically, covering a large spatial region centred on the zenith (Hocking et al., 2001). The typical output from these radars is a time series of two-dimensional (2-D) horizontal wind vectors covering a range of altitudes at a 1-hr cadence, although the motion of individual meteors can be identified at a much higher cadence. These horizontal wind vectors can be resolved into meridional and zonal components of the neutral wind. In contrast, the SuperDARN radars sound obliquely, with signals typically having elevation angles between ∼0° and 45°. Scattering profiles of individual meteor trails can be identified in very high-time resolution unprocessed SuperDARN data (Yukimatu and Tsutsumi, 2002; Tsutsumi et al., 2009), similar to the profiles measured by VHF meteor radars. However, the complex analysis required means that using such observations to study neutral wind variations over very long time intervals is presently impractical. The typical neutral wind product from the SuperDARN radars is a single time series of 2-D vectors using wind measurements averaged over a range of altitudes, also at a 1-hr cadence (Bristow et al., 1999). Although very suitable for studying low-frequency tidal fluctuations and planetary waves, this time resolution restricts the analysis of tides to the lowest frequency components.

In this paper we investigate whether measurements made at the standard (1-min) time resolution of the SuperDARN data set can provide additional and complementary understanding of tidal (and other) variations in the MLT. Here, we analyse meteor echoes from the Falkland Islands SuperDARN radar located at mid-latitudes in the southern hemisphere (Hibbins et al., 2011), using periodogram analyses. We propose that these higher-time resolution observations can provide complementary information that is unresolved in lower-time resolution data, and that they can help the development of models of tidal variations in the MLT.

The outline of the paper is as follows. Section 2 presents instrumental details about the SuperDARN radars as well as details of the data analysis methods used. Section 3 presents the results of a periodogram analysis of Falkland Islands SuperDARN radar data for the two-year epoch covering 2010 and 2011. Section 4 discusses these results, compares them with previous lower-resolution analyses, and speculates as to what the potential impact of higher-time resolution measurements might be. Section 5 summarises and concludes the paper.

## 2 Method

### 2.1 Instrumentation

SuperDARN (Greenwald et al., 1995; Chisham et al., 2007; Nishitani et al., 2019) is a network of $\sim$30 coherent scatter radars that transmit and receive radio signals in the HF band (specifically between 8 and 20 MHz), and were designed to study winds, waves and tides in the Earth's ionosphere and upper atmosphere. The SuperDARN radars are typically electronically-steerable, narrow-beam, phased-array radars (Greenwald et al., 1985). They trasmit multi-pulse sequences and the returned echoes are sampled and processed to produce multi-lag complex autocorrelation functions (ACFs) which are averaged over 3 s, and from which the signal-to-noise ratio (or power), line-of-sight Doppler velocity, and Doppler spectral width, can be estimated. The radars transmit signals over a range of oblique angles (typically $\sim$0° to 45° elevation), measuring echoes from backscatter targets at ranges $\sim$180 km to greater than $\sim$3000 km. They typically transmit in 16 to 24 narrow beam directions covering an azimuthal spread of greater than 50°. The raw SuperDARN data presented here were processed using the SuperDARN Radar Software Toolkit (RST), employing version 2.5 of the FitACF algorithm (doi:10.5281/zenodo.7467337).

At ranges $<\sim$400 km, the majority of SuperDARN backscatter targets are ionised meteor trails in the MLT, which occur over a range of altitudes from $\sim$75 to 125 km (Chisham and Freeman, 2013). Hall et al. (1997) first showed that the grainy near-range echoes typically observed by SuperDARN at these ranges are scattered from the ionisation trails of meteors ablating in the MLT. Individual meteor echoes are ubiquitous at these SuperDARN near ranges and are often characterised by a high signal-to-noise ratio, although the echo intensity usually appears to be random from one measurement to the next. Their Doppler spectra are very narrow, meaning that the measured velocity is well defined with low uncertainty (Hall et al., 1997). Consequently, each meteor echo provides an accurate estimation of the line-of-sight component of the neutral wind at the trail altitude. As discussed above, SuperDARN meteor echoes have been used to study tides in the MLT (van Caspel et al., 2020; Hibbins et al., 2019, 2011). In addition, they have also been used to study planetary waves (Stray et al., 2014, 2015) and to calibrate SuperDARN interferometers (Chisham and Freeman, 2013; Chisham, 2018; Chisham et al., 2021).

The Falkland Islands radar (FIR) (Hibbins et al., 2011) is one of only a few SuperDARN radars operating at mid-latitudes in the southern hemisphere, being located at 51.8°S and 59.0°W in geographic co-ordinates. At this longitude, the difference between Universal Time (UT) and Local Time (LT) is approximately 4 hours. The data analysed in this paper are from the first epoch of FIR operations that straddled the years 2010 and 2011. This choice of epoch is to allow comparison with the analysis of the lower-resolution data for the same epoch that was presented by Hibbins et al. (2011). Here, we only show the analysis of data from the geographic meridional beam of the radar (beam 6), and from range gate 4 (equivalent to a range of 315 km) (Chisham, 2023). (It is not possible to study meteor wind variations at high-time resolution in the zonal direction with the FIR radar, as there is no beam oriented in that direction.) The measured velocities are converted from the radial line-of-sight direction to the horizontal meridional direction by dividing by the cosine of the typical elevation angle for this range, as in Hall et al. (1997); we assume that the typical meteor height is 100 km for this purpose, as determined by Chisham and Freeman (2013). Similar to previous SuperDARN meteor wind studies we apply a minimum signal-to-noise ratio threshold of 3dB to remove poor and noisy data.

When using meteor echoes to measure neutral wind variations it is important to be aware of the potential for contamination
of the data set with other echoes, such as those originating from the ionospheric E-region and Polar Mesosphere Summer
Echoes (PMSE). E-region echoes can be particularly problematic as E-region scattering targets do not typically move under
the influence of the neutral wind. Hence, to briefly show the near-range echo populations typically observed by FIR, in figure 1
we present a simple statistical analysis of echoes from the first 20 range gates of FIR beam 6 (180-1035 km) for the month of
June 2010, which highlights the differences between various echo populations. Figure 1a presents the mean number of echoes
observed per hour ($\bar{N}$) for each range gate over the month. Figure 1b presents the mean point-to-point variation of the echo
power (or signal-to-noise ratio)($\overline{\Delta_P}$). Figure 1c presents the mean Doppler velocity ($\bar{V}$). Figure 1d presents the interquartile
range of the Doppler velocity distribution ($V_{IQR}$). In panels b-d, the statistics are assumed to be unreliable if $\bar{N}$ is 2 or less,
and these regions are greyed out in the figure.

Figure 1a shows that there are significant echo populations (orange) at the nearest ranges (where we expect meteor echoes
to occur), and at farther ranges (most likely sea scatter), for which $\bar{N} \sim$15, or higher. The maximum cadence of measurements
in a single SuperDARN range-beam cell during standard operations is one measurement every minute, which implies that a
maximum of 1440 meteor wind observations can potentially be made at every location each day. In reality, observations of
meteor echoes are sporadic and their number varies with time of day and time of year (Hall et al., 1997). The results presented
in fig.1a suggest around $\sim$15 echoes every hour in the morning hours, equivalent to $\sim$360 meteor wind observations at those
ranges every day. This number reduces significantly later in the day. Figure 3 of Hibbins et al. (2011) presented the seasonal and
diurnal variation of the monthly mean number of meteor echoes observed during the first year of FIR operations. During the
winter, there was evidence of a greater number of meteor echoes in the morning, similar to fig.1a, with the maximum occurring
around $\sim$0200 LT ($\sim$ 0600 UT). The occurrence of meteors was much reduced in the afternoon sector, again matching fig.1a.
In the summer, the overall meteor occurrence was reduced, with the maximum occurrence rates located around $\sim$0700 LT
($\sim$ 1100 UT).

Based on the observed statistical variations in all the panels of fig.1, we have identified three clear echo populations that
we have denoted A, B, and C. Population A extends for all times and for ranges $\sim$225-450 km, and shows the characteristics
expected of meteor echoes. These are: (1) Large number of echoes observed in the morning hours, reducing in the afternoon
(fig.1a). (2) Large point-to-point variation in echo power, symptomatic of the random change in echo intensity from one
measurement to the next (fig.1b). (3) Diurnal variation in Doppler velocity that matches the expected dominant semi-diurnal
tide in the neutral wind (fig.1c). We don't include the first range (180 km) in this population due to the potential contamination
by PMSE at very low ranges (Ogawa et al., 2003). However, it is possible that the scattering targets responsible for PMSE
do move with the neutral wind (Ogawa et al., 2013). Our chosen range for the analyses in this paper is in the centre of this
population (315 km - horizontal dotted line), where we might expect the contamination from the other populations to be at its
smallest.

Population B occurs mainly between $\sim$1300 and $\sim$2300 UT and for ranges $\sim$550-1050 km, and shows the characteristics
expected of sea scatter. These are: (1) Consistent power from one echo to the next, i.e., low point-to-point variation in echo
power (fig.1b). (2) Very low mean Doppler velocity ($\sim$0 km/s) as expected for a quasi-static target (fig.1c). (3) Very low

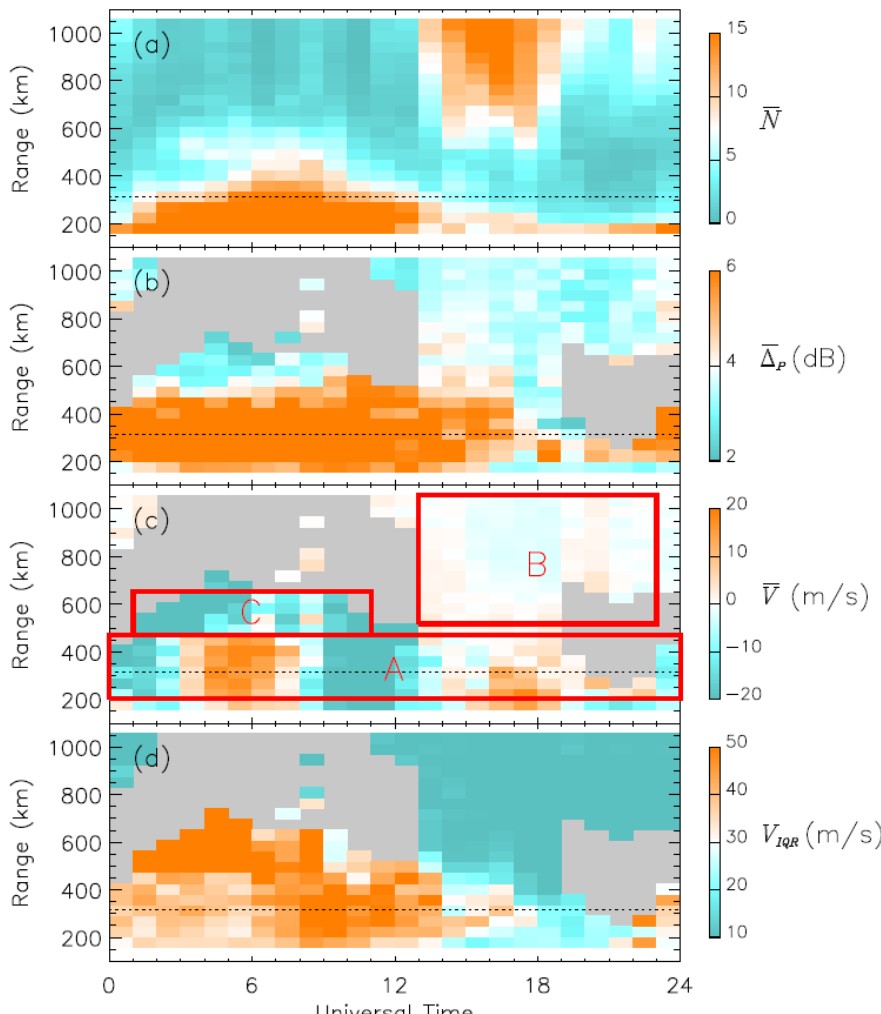

**Figure 1.** Echo statistics for the first 20 range gates (180-1035 km) measured by FIR beam 6, for the month of June, 2010. (a) Mean number of echoes observed per hour. (b) Mean point-to-point variation in echo power (in dB). (c) Mean Doppler velocity (in m/s). (d) Interquartile range of the Doppler velocity distribution (in m/s). The grey regions represent areas where there are not enough echoes to accurately estimate the statistics. The horizontal dotted line highlights the location of range gate 4 (315 km). The three red boxes (labelled A, B, and C) represent the locations of three distinct echo populations.

interquartile range of the velocity distribution indicating very little change in the observed Doppler velocity with time (fig.1d).
It is possible that these echoes form a contaminating factor to the meteor echoes between 1300-1900 UT, but this is likely to only be a factor at the farther ranges ($\sim$400+ km).

Population C appears smaller, extending from $\sim$0100 to $\sim$1100 UT and from $\sim$500-650 km in range, with a lower number of echoes per hour than seen in populations A and B. This population appears most likely to be a result of multiple short

intervals of E-region echoes. This would match the following observed features: (1) Consistent power from one echo to the next, i.e., low point-to-point variation in echo power (fig.1b). (2) High mean Doppler velocity that would be characteristic of ionospheric flow (fig.1c). (3) Very high interquartile range of the velocity distribution indicating large temporal fluctuations in the observed Doppler velocity (fig.1d). Due to the proximity of this E-region population with the meteor echo population there is the potential for contamination. To attempt to address this possibility, we also apply the selection criteria developed and used by Chisham and Freeman (2013) and Chisham (2018). Chisham and Freeman (2013) developed this method to reduce E-region contamination in near-range echoes measured by the Saskatoon SuperDARN radar, within the auroral zone, by understanding the differences in the probability distributions of the velocity error and spectral width error for the different echo types (as explained in detail in section 2.4 of that paper). Although not removing this E-region contamination completely, the proposed filtering reduced it significantly. However, this method may not be as effective at removing E-region echoes at mid-latitudes. Yakymenko et al. (2015) presented observations of E-region echoes at near ranges observed at mid-latitudes in the northern hemisphere; their observations suggested that the characteristics of E-region echoes at mid-latitudes may be different to those seen in the auroral zone. Consequently, we cannot be completely sure that all contaminating backscatter has been removed from the meteor echo data set. However, visual inspection of daily scatter plots from FIR indicates that the near ranges are overwhelmingly dominated by meteor echoes. We discuss the potential effects of any contaminating backscatter on the neutral wind tidal observations later in the paper.

In this paper we also present hourly-averaged SuperDARN neutral wind data, to show the comparison between the two data sets with different cadence. The method was first outlined by Bristow et al. (1999) and determines hourly mean velocity measurements in both the geographic meridional and zonal directions, combining raw velocity data from all available beam directions and the nearest ranges (180-405 km), by the application of a singular value decomposition (SVD) method (Hussey et al., 2000; Hibbins et al., 2007). Pre-processing requires that the raw input data points have a signal-to-noise ratio of greater than 3 dB and a line-of-sight Doppler velocity of less than 100 m/s. In addition, the pre-processing requires the velocity error of the input data to be less than 50 m/s, and the Doppler spectral width to be less than 25 m/s.

## 2.2 Time series analysis

For any particular range-beam cell, the high-resolution velocity time series data are typically unevenly sampled due to the non-continuous nature of the occurrence of meteors, and the requirement that the meteor echoes received match particular criteria, as discussed above. Figure 2 presents an example time series of the meridional wind estimates from FIR beam 6, range 4 for days 145 (26 May) through to 148 (29 May), 2010 (inclusive) as black square symbols. Low-frequency periodic variations (relating to tidal fluctuations) are clearly visible in this high-time resolution data, although there is a high level of point-to-point variability and 'noise' that introduces scatter in the neutral wind variations.

To place this data into context with the previous lower-resolution analysis of Hibbins et al. (2011), we compare our sample data set with the hourly-averaged meridional wind variation determined using the SVD method. These data are presented as the red square symbols and red solid line in fig.2. The low-frequency periodic variations in both data sets are very similar, as the

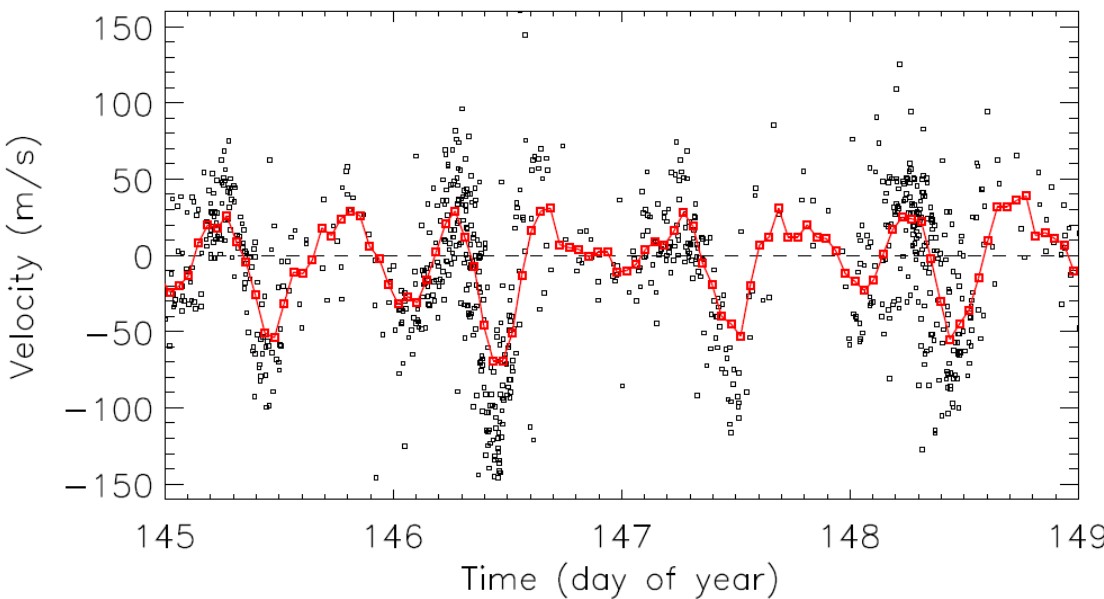

**Figure 2.** Temporal variation of the high-time resolution (1-minute cadence) neutral wind data from the FIR SuperDARN radar, beam 6, range gate 4 (315 km) (black symbols). The solid red line and symbols show the variation of the hourly-averaged FIR meridional wind data determined using the SVD method.

tidal components are dominant in both cases. However, as we will confirm later in this paper, more information about neutral wind fluctuations exists in the higher resolution data set.

The spectral analysis method that we use to analyse the neutral wind data is the normalised Lomb-Scargle periodogram
(Lomb, 1976; Scargle, 1982), which is extremely well-suited for analysing unevenly-sampled time series data. Periodogram analysis is often used to identify periodic signals hidden in 'noise', which makes it ideal for the analysis of tides in the neutral wind fluctuations. The periodograms have been determined using the definitions given by Scargle (1982) and Horne and Baliunas (1986). The analysis software uses the fast algorithm outlined by Press and Rybicki (1989).

Figure 3 presents the Lomb-Scargle periodogram (black solid line) for the 4-day interval of high resolution data presented
in fig.2. The frequency axis is presented in units of cycles/day which allows easy identification of the strength of the different tidal components: diurnal (1 cycle/day), semidiurnal (2), terdiurnal (3), quaddiurnal (4), etc. In this instance, the periodogram exhibits its largest peaks at 2 and 3 cycles/day, showing that the semidiurnal and terdiurnal tides are dominant at FIR at this time of year, as shown previously by the results of Hibbins et al. (2011). It is also clear that there are significant diurnal and quaddiurnal tidal components at this time.

The units on the power scale change relative to the number of input data points in an interval. Hence, the significance of a peak in the periodogram is best assessed by its relationship to prescribed significance levels. We make use of the False Alarm Probability (FAP) described by Scargle (1982) and Horne and Baliunas (1986) to define a significance level in each

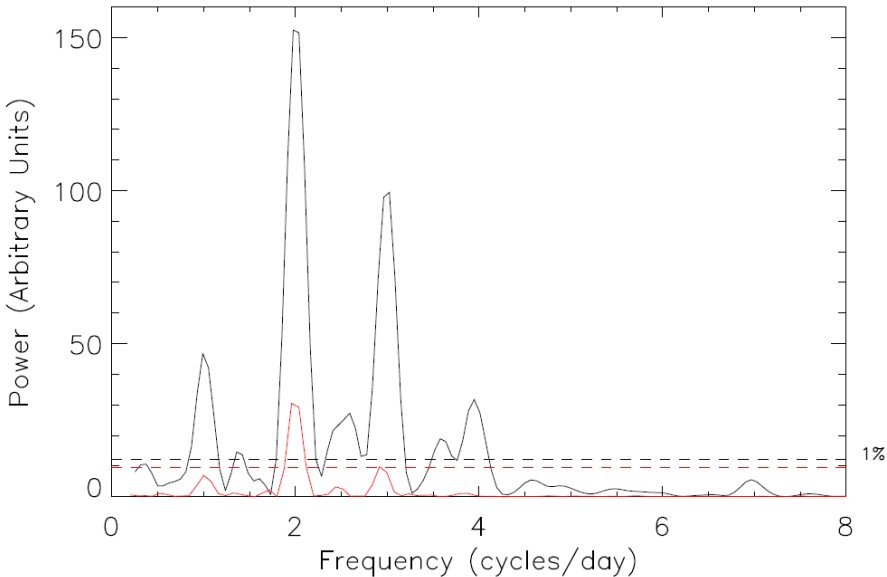

**Figure 3.** Normalised Lomb-Scargle periodogram of high-time resolution FIR meridional wind data from days 145 to 148, 2010, inclusive (solid black line). The horizontal black dashed line highlights the 1% FAP power level for these data. The solid red line presents the normalised Lomb-Scargle periodogram of the hourly-averaged FIR meridional wind data for the same interval. The horizontal red dashed line highlights the 1% FAP power level for these data.

periodogram, to assess the likelihood that any of the suspected periodic signals are real. The FAP represents a power level in the periodogram that will only be exceeded by chance a small fraction of time (given by a probability $p_0$). Here, we determine the FAP power level for $p_0$=0.01 (or 1%), and consequently assume that we have a 99% confidence in the periodic components with a power above this line. In order to correctly identify the power level for a particular FAP, it is important to normalise the periodogram by the total variance of the input data set, and not by an estimate of the noise derived either from the residuals after a periodic signal has been removed or from the uncertainty in the measurement, as might be the case for other methods. The black dashed horizontal line in the periodogram figures in this paper shows this 1% FAP power level. As the different numbers of points in each interval give different power scales in the periodogram, we compare the results from different periodograms by examining their power relative to this 'significance' level.

In the study of Hibbins et al. (2011), they identified the amplitude and phase of the tidal components in the data set by taking 4-day intervals of data and modelling them by performing nonlinear least-squares fits of a function comprising sine waves with periods of 48h (representing a 2-day wave), 24h (diurnal tide), 12h (semidiurnal tide) and 8h (terdiurnal tide). That approach restricted the spectral analysis to those chosen frequencies. However, Scargle (1982) and Horne and Baliunas (1986) showed that a periodogram analysis and the least-squares fitting of sine waves are equivalent ways of analysing a time series data set. In fig.3 we also present the periodogram for the hourly-averaged meridional wind data presented in fig.2 (red

solid line). This result is hence equivalent to the spectral analysis undertaken by Hibbins et al. (2011), but extends over a

wider range of frequencies. The red dashed horizontal line shows the associated 1% FAP power level, which is below that for

the higher resolution data. The peaks in the periodogram for the lower-resolution data are severely reduced, with the diurnal

peak not reaching significance, and the terdiurnal peak being barely significant at the 1% level. In particular, the strength of

the terdiurnal tide compared to the semidiurnal tide is smaller than is seen for the higher resolution data. These subtleties are

important if such results are going to influence modelling efforts. These differences show the effect of reducing the resolution

of the data set before applying any spectral analysis technique. Preprocessing of data, such as linear interpolation to an even

spacing, can result in the loss of information. Rebinning unevenly sampled data to equally spaced bins, and then calculating

a conventional periodogram, may alter the frequency and significance of any periodic signals (Horne and Baliunas, 1986). In

addition, any consequent spectral analysis can completely miss or reduce the size of important peaks, e.g., there is no sign of

any quaddiurnal tide in the periodogram of the hourly-averaged data. This suggests that higher resolution analysis is needed

to properly assess the importance of higher-frequency tidal components such as the quaddiurnal tidal component and other

significant frequencies that might occur outside of the fixed frequencies used in least-squares fits.

Rather than use 4-day intervals (as in Hibbins et al., 2011), in this study we select 10-day intervals for our standard peri-

odogram analysis. This results in a higher frequency resolution in the periodograms, making the peaks sharper and more clearly

defined. Figure 4 presents the periodogram for a 10-day interval that includes the 4-day interval studied above (Days 140 to

149, 2010, inclusive). In this case the periodogram shows the same clear diurnal, semidiurnal, terdiurnal, and quaddiurnal tidal

components as in fig.3, but the peaks are much sharper and extend to higher power levels above the 1% FAP power. In fig.4

(and in most of the subsequent periodograms), there are very small but 'significant' peaks close to the main tidal peaks that are

likely to be a result of spectral leakage, as discussed in detail by Horne and Baliunas (1986). These most likely occur due to

the use of a finite data window. It is important to be aware of this spectral leakage when interpreting the periodograms.

## 3   Results

To provide an overview of the complete epoch of FIR operations in 2010 and 2011, we have produced a two-dimensional

(2-D) periodogram of the epoch. A series of periodograms were determined covering 10-day windows, stepping in 2-day

intervals across the epoch. The resulting 2-D periodogram is presented as a contour plot in fig.5a. Here, the darker regions

highlight where the power is significant (greater than the 1% FAP power level). To help with the visual identification of

seasonal variations in the data, winter and summer are also marked on the figure as blue and red blocked regions, respectively.

Figure 5b presents the number of data points ($N_0$) that were used to compile each periodogram. This helps to identify

intervals where the quality of the periodograms may be affected by a reduction in the number of input data points. In particular

it highlights a period of downtime in the radar operations in early April 2010, as shown previously in the data coverage plot of

Hibbins et al. (2011), fig.2.

Periodogram peaks related to the tidal components in the data set are clearly visible as persistent horizontal bands of high

power across the epoch. This provides a visual picture of the temporal variation of the strength of the diurnal, semidiurnal,

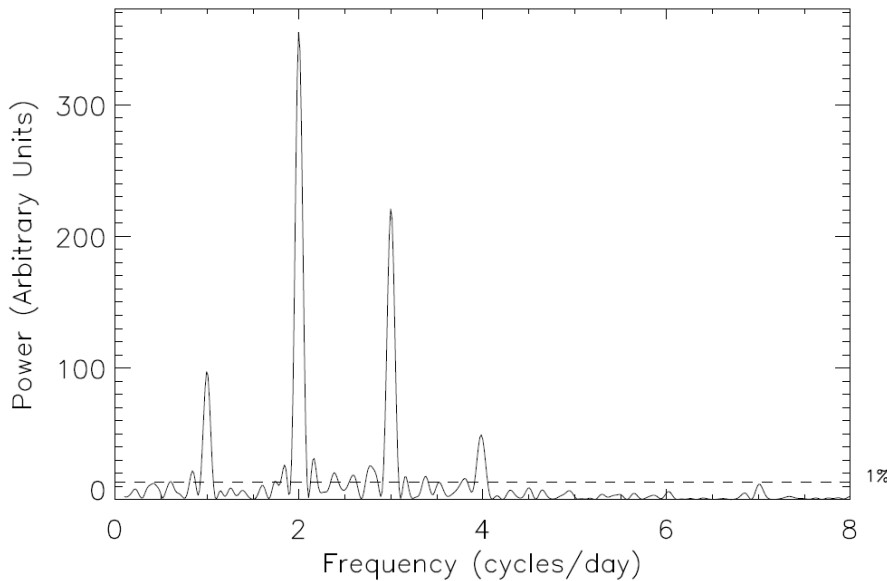

**Figure 4.** Normalised Lomb-Scargle periodogram of the high-time resolution FIR meridional wind data from days 140 to 149, 2010, inclusive. The horizontal black dashed line highlights the 1% FAP power level.

terdiurnal, and even higher frequency tidal components. There are also persistent bands of power at other frequencies at particular times of year.

Of the tidal components, the semidiurnal and terdiurnal components display the greatest power, and are the most persistent throughout the epoch. These tides are strong from around the autumnal equinox (March) until after the vernal equinox (October). They are particularly strong in the months around the beginning of winter, from April to July. There are a few dropouts in the tidal power during this time, some of which appear correlated with reductions in data coverage (as shown in fig.5b). However, the power in these tidal components drops out almost completely from the start of the summer (November) until close to the autumnal equinox (March), with significant peaks at these frequencies only appearing sporadically for small intervals during this time.

The diurnal tidal component persists for similar intervals as the semidiurnal and terdiurnal tides across the epoch, but the power in this component is reduced in comparison. The quaddiurnal tidal component is not as persistent as the other components, appearing mainly in the months around the beginning of winter (April and May) and around the vernal equinox (September); the power in this component is typically similar to that seen in the diurnal component. Significant power at even higher frequency tidal components can be seen for small intervals of time across the epoch, especially near the equinoxes.

Figure 6 presents an example 10-day periodogram from within the interval characterised by very strong tidal variations in 2010 (days 110 to 119 inclusive). As is the case throughout the epoch, the semidiurnal tidal component is the largest, although other tidal components (diurnal, terdiurnal, and quaddiurnal) are also highly significant, with the terdiurnal component being

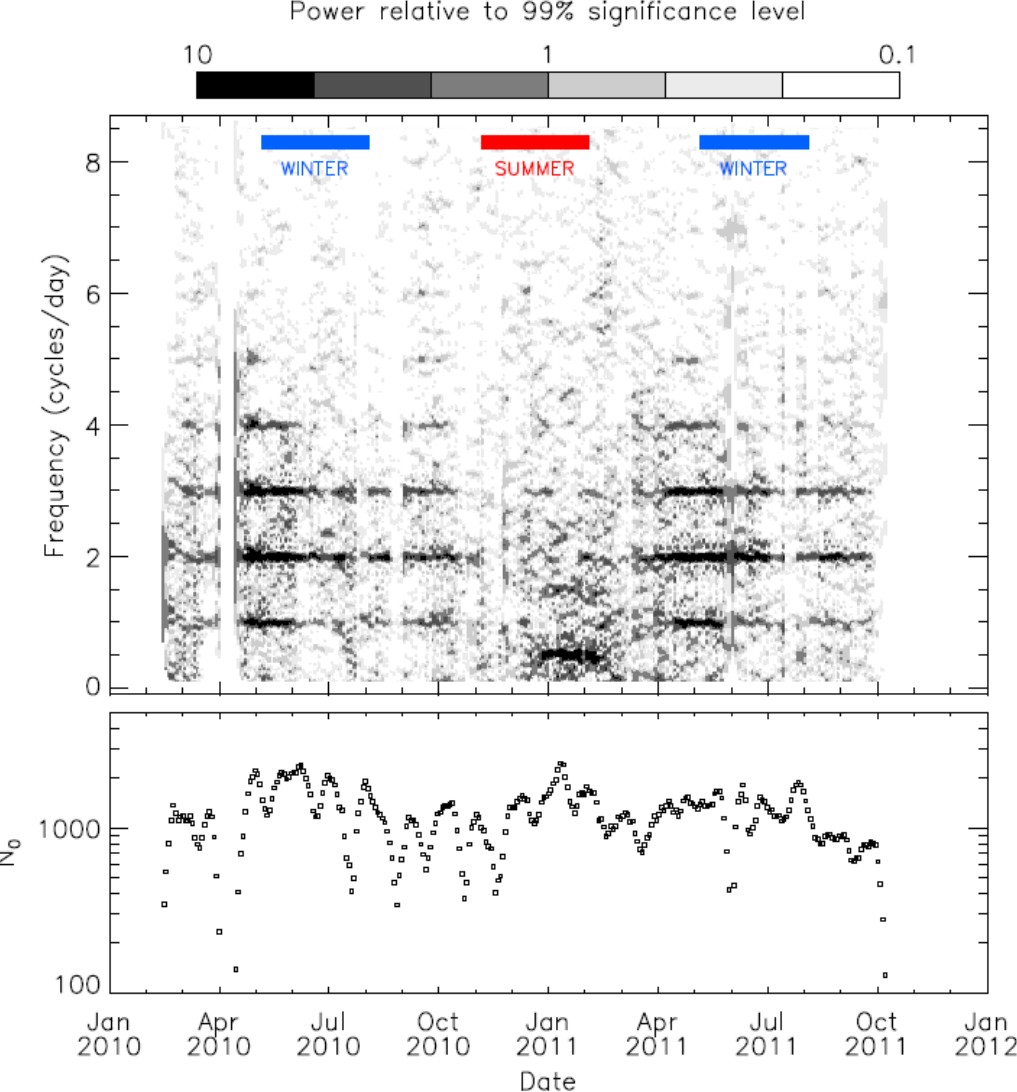

**Figure 5.** (a) Two-dimensional periodogram of the FIR neutral wind data from beam 6, range gate 4 (315 km), for the first epoch of FIR operations covering 2010 and 2011. The two-dimensional periodogram is compiled from a series of 10-day periodograms stepping in 2-day intervals. The periodogram power levels are presented relative to the 1% FAP power level. The darker regions represent the regions of higher power. The extents of summer and winter across this epoch are highlighted by the red and blue blocked regions, respectively. (b) The number of time series data points ($N_0$) that that were used in the compilation of each 10-day periodogram, separated at 2-day intervals.

almost as large as the semidiurnal component. There are also smaller peaks either side of each of the main peaks that appear to

be the result of spectral leakage. The existence of significant peaks resulting from spectral leakage is a feature of much of the epoch. These spectral leakage peaks can be clearly seen in fig.4a as dark spots either side of the main tidal bands.

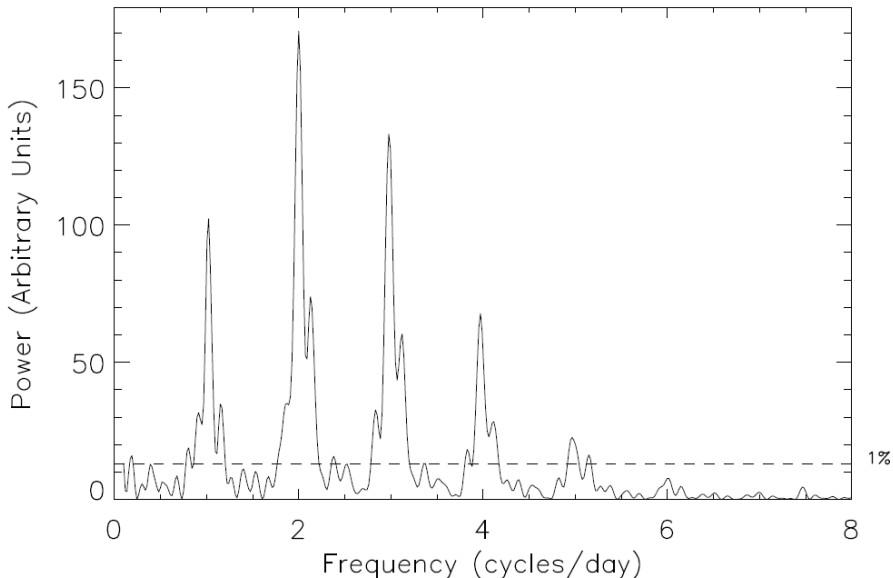

**Figure 6.** Normalised Lomb-Scargle periodogram of the high-time resolution FIR meridional wind data from days 110 to 119, 2010, inclusive. The horizontal black dashed line highlights the 1% FAP power level.

In fig.6 there is also a significant tidal component at 5 cycles/day (which we term pentadiurnal). Significant tidal components at this, and higher harmonics, are not unusual throughout the measurement epoch, although they tend to persist for only small intervals of time. Figure 7 presents a 10-day periodogram from very close to the vernal equinox (days 255 to 264, 2010, inclusive). Here, in addition to the five significant tidal components visible in fig.6, there are also significant tidal components visible at 6 cycles/day (hexadiurnal), and 7 cycles/day (heptadiurnal). Observations of these higher harmonics have rarely been reported before.

As discussed above, all the tidal components appear to completely disappear ∼2 months before the summer solstice, returning just before the autumnal equinox. During part of this time the meridional winds are dominated by a quasi-two-day wave (QTDW) (around 0.5 cycles/day) that starts just before the summer solstice and lasts for about 2 months. Figure 8 presents a zoomed-in version of the 2-d periodogram for the summer interval encompassing the QTDW. This shows that the QTDW appears to be accompanied by another persistent fluctuation with a period of ∼16-hr (1.5 cycles/day). However, the frequency of both these fluctuations does appear to change slightly with time in contrast to the tidal components at other times of the year, for which the frequences are unvarying. Also within this interval are other significant peaks of shorter duration where the frequency of the fluctuations changes with time. These often occur in fig.8 as rising or falling bands of power, covering a range of frequencies, and lasting for ∼10-20 days each. There are also instances of short-term tidal variations across the interval; the

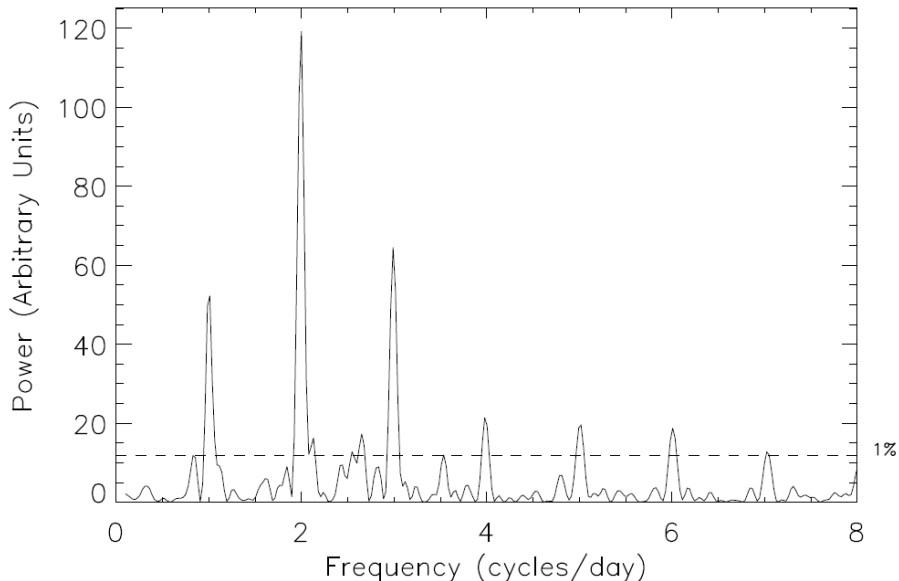

**Figure 7.** Normalised Lomb-Scargle periodogram of the high-time resolution FIR meridional wind data from days 255 to 264, 2010, inclusive. The horizontal black dashed line highlights the 1% FAP power level.

diurnal tide is present for short intervals (∼10 days) both before and after the extent of the QTDW. Similarly, the semidiurnal and terdiurnal tides are present for short intervals both before and after the extent of the 16-hr wave.

Figure 9 presents an example 10-day periodogram from the centre of this interval (days 5 to 14, 2011, inclusive). First, it is clear that this periodogram is devoid of any tidal components. It is dominated by the peak relating to the QTDW (close to 0.5 cycles/day); the peak at this time is actually ∼0.54 cycles/day, which corresponds to a period of ∼44.6h. There are two other clear peaks in fig.9. The first is that seen in the persistent band around 1.5 cycles/day, corresponding to a period of ∼16 h. The second, around 2.2 cycles/day is part of a short band of increasing frequency that lasts for about 15 days.

## 4 Discussion

In this paper we have shown that a Lomb Scargle periodogram analysis of high-resolution SuperDARN neutral wind data provides complementary information to the standard analysis of hourly-averaged SuperDARN meteor wind data that has been employed in most SuperDARN neutral wind studies to date. A combination of both these different analysis methods has the potential to provide a better understanding of tidal and other low frequency fluctuations in the MLT, potentially improving opportunities for data assimilation and the empirical inputs to atmospheric models. Here, we discuss comparisons with previous observations, the limitations of the instrumentation and the analysis method, and the new opportunities that this higher-resolution analysis may afford.

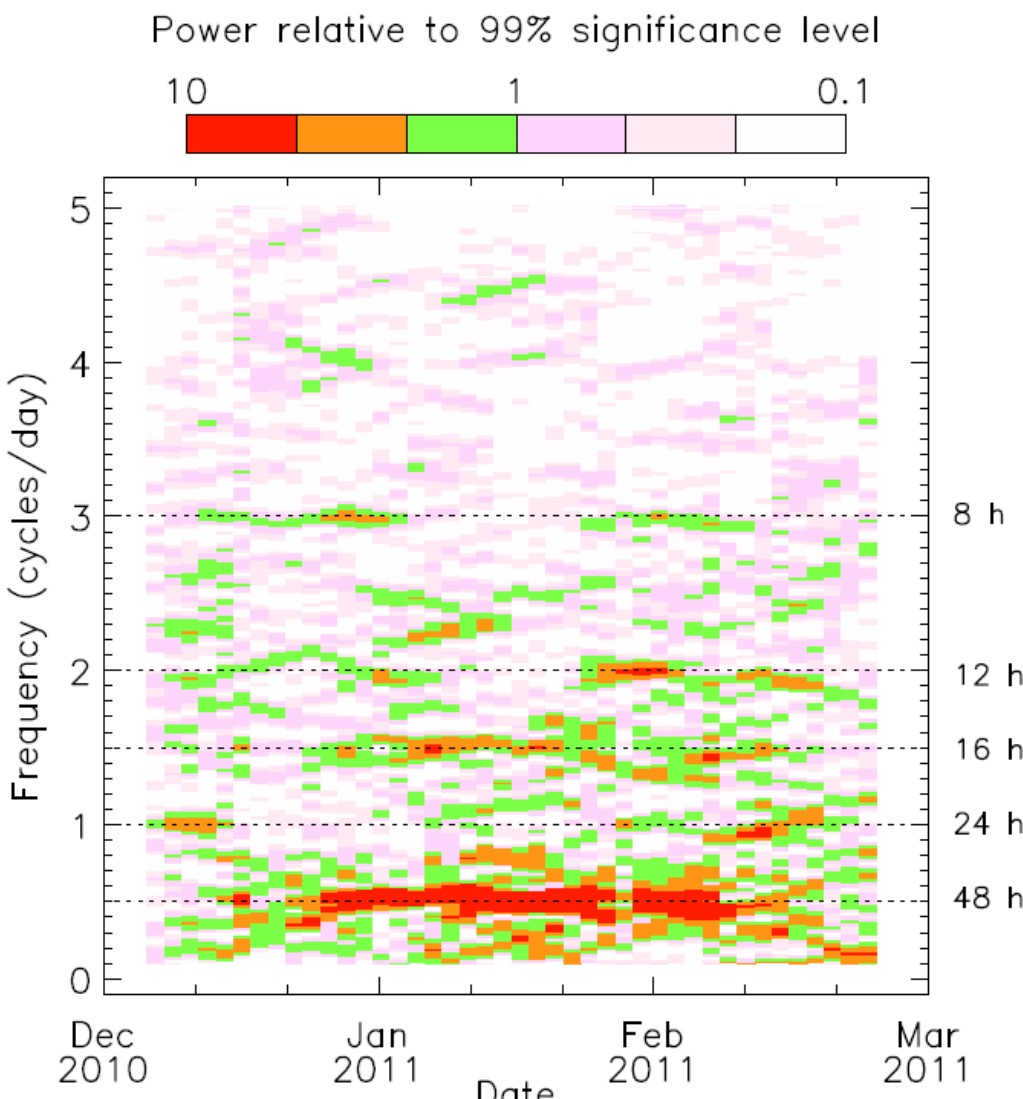

**Figure 8.** Two-dimensional periodogram of the FIR neutral wind data from beam 6, range gate 4 (315 km), for the interval Dec 2010 to Mar 2011. This is a zoomed in version of figure 5 to show the quasi-two-day wave and associated variations in more detail. The two-dimensional periodogram is compiled from a series of 10-day periodograms stepping in 2-day intervals. The periodogram power levels are presented relative to the 1% FAP power level.

## 4.1 The potential impact of non-meteor echo contamination on the tidal signals

Before discussing the scientific results, we first consider the impact that any possible contamination from non-meteor echoes may have on our results. Typically, there are multiple potential scattering targets within the fields-of-view of the SuperDARN

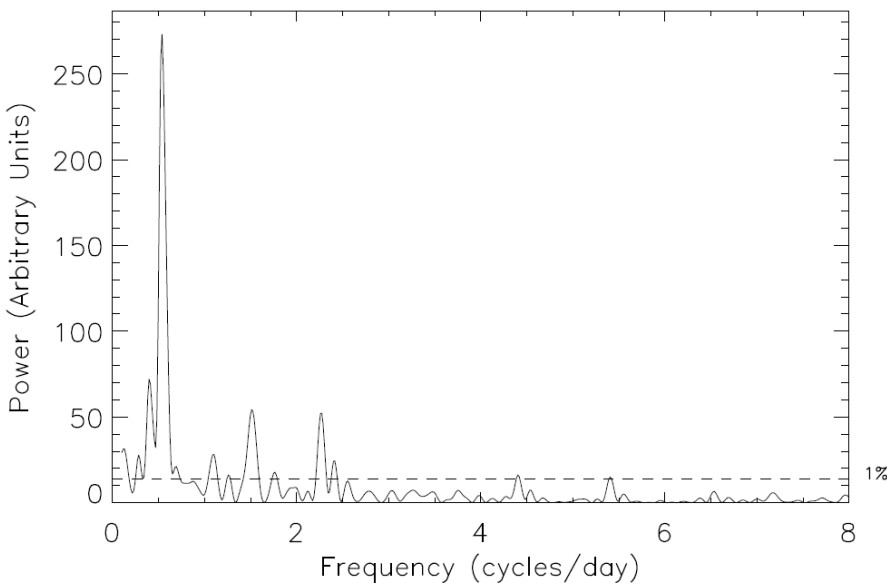

**Figure 9.** Normalised Lomb-Scargle periodogram of the high-time resolution FIR meridional wind data from days 5 to 14, 2011, inclusive. The horizontal black dashed line highlights the 1% FAP power level.

radars. Hence, there has always been a problem with contamination of the measured signal by unwanted echoes. This is most
280 pertinent at farther ranges where F-region scatter is often mixed with ground or sea scatter. A large amount of effort has been expended over the last few decades to develop algorithms that can automatically filter SuperDARN data to remove ground and sea scatter. Even given this concerted effort, the methods to separate different types of scatter are imperfect, and contaminating backscatter will often remain.

The same issues regarding distinguishing between different echo populations exist in near-range studies, particularly between
285 meteor echoes and ionospheric E-region scatter. The overlap of different echo populations at near ranges has not been given as much intensive thought in the community, and varying disparate methods for removing unwanted scatter have been applied in different studies. A reliable method for separating echo populations at near ranges would be of great benefit for the SuperDARN community and should be a topic of future study.

In most studies of near-range scatter the methods used to remove unwanted contaminating backscatter have generally not
been fully proven. Hence, the main question that needs to be asked is 'what is the effect of any remaining contamination on the signals that we wish to study?'. This study concentrates on tidal signals and other large-scale periodic neutral wind oscillations apparent in meteor echoes. Any contaminating echoes (either E-region or sea scatter) are likely to have line-of-sight velocities which are not components of the motion of the neutral wind. The impact of such contamination would be to reduce the amplitude of the tidal signals that are observed. The fact that we still see strong tidal signals for much of the year
suggests that this contamination is typically minimal. However, there are instances in the data set where the tidal signals drop out. Sometimes, this can be attributed to reductions in data coverage, but there may be times when contaminating backscatter

is an issue. During the summer months the tidal signals drop out almost completely. However, the presence of other persistent periodic signals during this interval suggests that this is not a consequence of contaminating backscatter. The full effects of contaminating backscatter in near-range meteor echoes needs to be a topic of future study.

## 4.2 The temporal variation of the main tidal components

The tidal components at FIR during this epoch were first studied by Hibbins et al. (2011) using the hourly-averaged SVD neutral wind data products. The main results of Hibbins et al. (2011) can be summarised as follows: (1) The semi-diurnal tide was the dominant periodic component in the winter months; (2) Large amplitude bursts of QTDW activity were observed in the summer months. Here, we compare our observations of the main tidal components with those of Hibbins et al. (2011).

Hibbins et al. (2011) observed that the diurnal (24h) tide was weak throughout the year. Our results show the diurnal tide as relatively strong and persistent, although weaker in magnitude than the semi-diurnal (12h) and terdiurnal (8h) tides. Our results also suggest that this tidal component drops sharply to zero in the summer months.

Hibbins et al. (2011) identified the semidiurnal (12h) tide as the strongest component, and that it was strongest in May, with a secondary maximum in late winter. We also see the semidiurnal tide as the strongest component, with the highest power in April, May, and June, but our results suggest that any drop in power during the winter may result from reduced meteor occurrence frequency. These drops in power may also occur as a result of non-meteor echo contamination in the data set. However, our results suggest that the tidal power in this component is persistent throughout much of the year before dropping sharply to zero for the summer months, similar to the diurnal component.

The results of Hibbins et al. (2011) suggested that the amplitude variation of the terdiurnal tide during the year was similar in shape to that of the semidiurnal component, but with about a third of the amplitude. We observe much stronger terdiurnal components than Hibbins et al. (2011), with the amplitude often approaching similar levels to the semidiurnal tidal amplitude. This is possibly the largest difference between the results from the two methods. Indeed, the reduction in the power of the tidal components with increasing frequency was clear in the periodogram comparison in fig.3.

The main reason for the reduced overall power in the tidal components using the SVD method, and particularly the increased reduction with increasing frequency is the effect of averaging the data over hourly windows in that method. The change of cadence in particular reduces the power in the higher frequency components. Another reason for the reduction in power in the SVD method may be the criterion that measurements with line-of-sight Doppler velocities greater than 100 m/s are removed in the pre-processing (Hussey et al., 2000; Hibbins et al., 2007). The removal of the higher velocity neutral wind measurements will have resulted in a significant reduction in the averaged wave amplitudes. This criterion was originally introduced as there was an expectation that neutral wind velocities are typically less than ∼100 m/s, and that higher velocities would imply scatter from E-region irregularities. This appears quite a simplistic method for removing potential E-region echoes; it can be seen in fig.2 that the high-time resolution horizontal velocities can extend to ∼150 m/s and possibly higher at the peaks of the semi-diurnal tide, whereas the red curve in fig.2 does not reach anywhere near the extremes observed in the higher resolution data. (It should be noted that Hall et al. (1997) suggested a velocity threshold of 200 m/s). Improving the algorithms for removing potential E-region scatter from the data set may improve the results obtained using the SVD methodology.

## 4.3 The occurrence of higher-frequency tidal harmonics

To date, observational studies of solar tides have mainly focussed on the low-frequency harmonics of the solar day, concentrating predominantly on the diurnal, semidiurnal and terdiurnal tides discussed above. The study of Hibbins et al. (2011) was restricted to the analysis of tidal components up to the terdiurnal tide, due partly to the least-squares fitting employed to determine the tidal amplitudes and phases, and partly to the temporal resolution of the data set. Consequently, higher frequency components are typically not observed by traditional SuperDARN, and other, meteor echo analyses. Although not often observed, higher harmonics do occur; solar heating of the atmosphere occurs in an approximate square wave profile and so is rich in harmonics. Here, we have shown that there are instances when there is significant power in higher-frequency tidal components, particularly in the quaddiurnal tide.

The quaddiurnal tide (often termed the quarterdiurnal tide) has had increasing recent attention, being observed both from space (Azeem et al., 2016; Liu et al., 2015) and from the ground (Smith et al., 2004; Jacobi et al., 2017; Gong et al., 2018; Guharay et al., 2018). Most studies show that the occurrence of the quaddiurnal tide is highest in the winter months (Smith et al., 2004; Liu et al., 2015; Jacobi et al., 2017). This is a slight contrast to the observations presented here, where the quaddiurnal tide is seen to be strongest between the autumnal equinox and the winter solstice, as well as around the vernal equinox, although a similar variation was seen in the monthly means of the quaddiurnal tide by Gong et al. (2021). They also showed that the tide enhanced during a Sudden Stratospheric Warming (SSW) event. Observations of the quaddiurnal tide occurrence over multiple years would help to resolve the differences between these earlier observations.

The amplitude of the quaddiurnal tide has been shown to increase with increasing altitude (Liu et al., 2015; Jacobi et al., 2017; Guharay et al., 2018; Gong et al., 2021), so much so that at thermospheric heights of ∼300 km the quaddiurnal component is as important as the diurnal and semidiurnal tides (Gong et al., 2018). It is possible that the significant quaddiurnal tides that we see in this paper are partially a consequence of the SuperDARN HF neutral wind observations being centred around 100 km, in contrast to the VHF meteor radar observations which are typically centred around 85 to 95 km. The vertical wavelengths of quaddiurnal tides are often observed to be very long, >∼100 km (Azeem et al., 2016; Jacobi et al., 2017), particularly in the winter months (Jacobi et al., 2017). The consequent small change in wave phase with altitude means that the observational nature of the SuperDARN meteor echoes, i.e., the fact that they are measured or averaged over a range of altitudes, will not result in a considerable reduction in the observed quaddiurnal tide amplitude as a result of phase mixing. Given our observations in this paper, and the global distribution of SuperDARN radars, it seems very likely that high-resolution SuperDARN data could be used to perform a detailed global study of the quaddiurnal tide.

Observations of higher harmonics than the quaddiurnal tide are very rare. Hedlin et al. (2018) observed harmonics up to the 10th harmonic of the solar day in surface pressure variations. At MLT altitudes, He et al. (2020) observed the first 6 solar tidal harmonics in ground-based radar observations. These harmonics all had peaks above the 1% FAP level in a Lomb-Scargle periodogram of almost three months of data covering northern hemisphere winter. They also showed that the higher harmonics were quenched after the onset of a SSW event. Here, we observed increased harmonic components near to equinox. It is

possible that the conditions of symmetric day and night across the globe at equinox is conducive for the persistence of multiple solar day harmonics. Future work using multiple years of data from multiple stations will help to investigate this further.

In general, our observations show that it is possible to identify these higher tidal harmonics in shorter intervals of data when using higher-time resolution data. This should make it easier to study this type of behaviour in the future and provides the potential to gain increased knowledge of high frequency tidal harmonics that can contribute toward the improvement of models.

## 4.4 The quasi-two-day wave (QTDW)

During the southern hemisphere summer, when the tidal modes are absent in the FIR observations, we see clear evidence of a persistent QTDW, which was first observed during this interval by Hibbins et al. (2011). The QTDW is a large amplitude global-scale wave that is regularly observed in the MLT (Craig et al., 1980; Salby, 1981; Wu et al., 1993); these waves have been seen to attain amplitudes $>\sim$60 m/s (Wu et al., 1993; Hibbins et al., 2011). It is a normal mode of the Earth's atmosphere that typically occurs in short intense bursts of activity, often following the summer solstice (Wu et al., 1993; Limpasuvan et al., 2005), and with enhanced amplitudes in summer (Lima et al., 2004). The duration of the wave can range from several days to intervals lasting more than a month (Salby, 1981). Our observation of the quasi-two-day wave appears exceptionally long, lasting for $\sim$2 months (see figure 8).

The QTDW has been identified and studied previously using SuperDARN radar data (Bristow et al., 1999; Malinga and Ruohoniemi, 2007; Hibbins et al., 2011), although all these studies employed hourly-averaged data. Malinga and Ruohoniemi (2007) used a longitudinal chain of SuperDARN radars covering $\sim$180° longitude to study the longitudinal behaviour of the QTDW, and deduce its zonal propagation characteristics. The global coverage of radars within the SuperDARN network makes them ideal for studies of this type. These SuperDARN studies also showed a tendency for the mean wave period to vary by several hours over an interval, possibly due to the superposition of different horizontal wavenumber modes (Malinga and Ruohoniemi, 2007). For the wave presented here, Hibbins et al. (2011) suggested that the period varied from $\sim$46.5h early in the interval to $\sim$51h later in the interval. Figure 8 shows a similar variation with the periodogram peak of the QTDW being of a lower period than 48-h at the start of the interval, and a higher period at the end. Meteor radar observations of the QTDW have shown that the period of the wave is typically more than 2 days, often around 2.1 days ($\sim$50 h) (Salby, 1981).

The major advantage of the high-time resolution SuperDARN observations that we present here is the ability to study higher frequency variations in greater detail. Our observations presented in fig.8 show the existence of a persistent 16-hr wave coincident with the QTDW, which was not identified in the lower-resolution observations of Hibbins et al. (2011), due to the use of fixed frequencies in their least squares fit. It has been proposed that the 16-h wave is a secondary wave resulting from nonlinear interactions between the QTDW and the diurnal tide (Teitelbaum and Vial, 1991; Pancheva, 2006; Nguyen et al., 2016; Lieberman et al., 2017). It has previously been observed concurrently with the QTDW during summer, although it is seen to maximise at lower altitudes than the QTDW, typically below $\sim$90 km (Manson and Meek, 1990).

Our observations also provide evidence of even higher frequency variations during the QTDW epoch that are not so regularly observed. We see waves with rising and falling frequency at higher frequencies, concurrent with the QTDW and the 16-h wave.

Some previous studies have observed a 9.6 h period wave (2.5 cycles/day) (Manson and Meek, 1990; Pancheva, 2006; Nguyen et al., 2016), although it has sometimes only been possible to identify this wave in aliased measurements or in reanalysis models (Nguyen et al., 2016). Our results (fig.8) clearly show this period of oscillation close to the start of the 16-hr component. However, it reduces in frequency until it merges with another frequency component at $\sim$2.3 cycles/day ($\sim$10.4 h periodicity). There is also evidence of an even higher significant frequency component at $\sim$4.5 cycles/day ($\sim$5.3 h periodicity) at this time. These higher frequency variations are typically short-lived ($\sim$10-20 days) and change systematically with time. It may be that they can provide important information about non-linear wave interactions at the time of the QTDW. The ability to study these higher frequency variations during the QTDW is a major benefit of this higher-resolution data set. Future work using multiple longitudinally-spaced SuperDARN radars will be important for studying the zonal structure of these higher frequency components.

## 4.5  The limitations and opportunities of SuperDARN high-time resolution analysis

Although the use of high-time resolution SuperDARN neutral wind measurements provides new observation capabilities, there are obvious limitations to this analysis which mean that it is best employed in combination with other, more established analysis methods.

One limitation, which is true of SuperDARN neutral wind observations at all temporal resolutions, is the limited height resolution of the SuperDARN tidal observations. Most of the SuperDARN radars are equipped with interferometers that allow the height of meteor echoes to be determined. However, there are not typically the large number of echoes available to provide a height profile of neutral wind variations, as meteor radars do. The height of meteor echoes is variable from one echo to the next, so the scatter does not come consistently from one altitude. Although early studies suggested that the average of SuperDARN meteor echo heights was around 95 km (Hall et al., 1997; Hussey et al., 2000), they have latterly been shown to be distributed over a range of altitudes, approximately 75-125 km, centred on 102-103 km, with a full-width at half maximum of 25-35 km (Chisham and Freeman, 2013; Chisham, 2018). Hence, the height range (and peak occurrence altitude) of meteors measured by HF radars is different to that of meteor radars that sound in the VHF range, due to the way that the meteor echo height ceiling effect varies with frequency (Thomas et al., 1998).

One consequence of combining observations from a range of altitudes is the mixing of phases in the wave and tidal observations, as they have finite vertical wavelengths. As the wave phases vary with altitude, combining observations can lead to the partial destructive interference of any wave, reducing the overall observed amplitude, and smearing the waveforms. Consequently, it is challenging for SuperDARN radars to resolve waves with small vertical wavelengths (<10km), such as gravity waves, even with accurate interferometer measurements. However, the semidiurnal, terdiurnal, and quaddiurnal tides often have vertical wavelengths of $\sim$100 km in the MLT (Yuan et al., 2008; Chapman and Lindzen, 2012; Azeem et al., 2016). Hence, tides with vertical wavelengths of this size are more easily resolved in SuperDARN observations, although often the amplitude of the waves can be reduced by $\sim$20% (Hibbins et al., 2011). Future high-time resolution analyses with the inclusion of interferometer measurements of echo height will help to explore the potential capabilities for improved height resolution.

Another limitation of the high-time resolution SuperDARN neutral wind analysis, in contrast to the standard SuperDARN SVD or meteor radar analyses, is the lack of a determination of the full horizontal velocity vector, which allows determination of both the zonal and meridional components. Those methods combine and average the measured neutral wind velocities from a range of different directions to resolve both components of the wind. In contrast, the high-time resolution method using a single beam that is presented here provides information about a single component only. However, we would argue that this analysis provides a better representation of the meridional wind component. Both components determined using the SVD method include the influence of noise and fluctuations from multiple velocity measurements over a range of directions which will affect the accuracy of the measurement. This is in addition to the reduced amplitude of the lower frequency variations that occurs as a result of the hourly averaging. Hence, we would argue that the purer representation of the meridional wind variation provided by the high-time resolution data provides additional opportunities for the analyses of different aspects of the observed tidal variations, such as the comparative intensities of the different tidal components, as discussed earlier.

The present analysis used data from a single observing station only. The SVD method provides information about the phase of the different tidal components that is not available from the single-station high-time resolution data. By combining phase measurements from longitudinally-separated radars, it is possible to determine whether the tidal variations relate to migrating or non-migrating tides (van Caspel et al., 2020). Future high-time resolution analyses need to concentrate on similar multiple observations from longitudinally-separated SuperDARN radars to assess how to best estimate the longitudinal phase variation between observations.

The high-time resolution nature of the SuperDARN neutral wind variations, as presented here, opens up opportunities for the analysis of higher-frequency variations, as shown by the observation of the high frequency tidal components and the higher frequency variations at the time of the QTDW. Given the very narrow spectra of meteor echoes, the velocities of these echoes are very well defined, with very low uncertainty ($< \sim 1$ m/s). Consequently, what appear to be (and are often interpreted as) instrumental 'noise' in high-time resolution time series (such as that presented in fig.2), are likely to be true neutral wind variations. Although some of this 'noise' may be a result of observing tidal variations from meteor echoes at different altitudes, most of these very small-scale, high frequency velocity variations are likely to be a signal of atmospheric gravity waves and/or turbulence.

As already discussed, it is difficult to resolve gravity wave fluctuations in SuperDARN data due to their small vertical wavelengths. However, it may be possible to use this data set to estimate what proportion of overall neutral wind fluctuations are due to tides, and what is due to gravity wave and turbulent fluctuations. Periodic (tidal) and the residual aperiodic components of the meridional neutral wind variation can be separated using median filtering methods such as nonlinear data smoothing (e.g., Velleman, 1977). These aperiodic components represent the signatures of gravity waves and turbulence, which play an important role in energy transfer and dissipation in the MLT. It may also be possible, using structure function analyses, to evaluate the nature of these aperiodic fluctuations, whether they are structured or turbulent.

# 5 Conclusions

High-time resolution meteor scatter measurements have been underused in previous studies of winds, waves, and tides in
the MLT region. The analysis in this paper shows that high-time resolution SuperDARN observations contain much more
information about fluctuations in the neutral wind than has previously been utilised. Using Lomb-Scargle periodogram analysis
of high-time resolution meteor wind data from the MLT allows the determination of the temporal variation of the magnitude
of a wide range of atmospheric tidal components and other large-scale oscillations, like the QTDW. In particular, it better
identifies the strength of higher frequency tidal components and their variations with time. Hence, this high-time resolution
periodogram analysis provides a complementary analysis method to the standard SuperDARN neutral wind analyses used in
previous studies. A better understanding of the variations in amplitude between different tidal components will help improve the
understanding of the physical mechanisms driving the different components, and will improve inputs to empirical atmospheric
models. This analysis may also open up opportunities for studying planetary waves, and gravity wave activity and turbulence
in the MLT.

*Code and data availability.* The processed FIR neutral wind data used in this paper are freely available at https://doi.org/10.5285/05fd4030-
9c22-4bfe-91c2-0f63ac376ec1 (Chisham, 2023). The raw SuperDARN data for years 2010 and 2011, which include all the raw FIR data, are
freely available through the BAS SuperDARN data mirror (https://www.bas.ac.uk/project/superdarn). The raw SuperDARN data were pro-
cessed using the SuperDARN Radar Software Toolkit (RST), employing version 2.5 of the FitACF algorithm (doi:10.5281/zenodo.7467337).
The IDL software used to produce the periodograms can be found at http://astro.uni-tuebingen.de/software/idl/aitlib/timing/scargle.pro, and
the documentation for this software is available at http://astro.uni-tuebingen.de/software/idl/aitlib/timing/scargle.html

*Author contributions.* GC had the idea for the analysis, wrote the analysis software, analysed the data, and wrote the initial draft of the paper.
GC and AJK worked on the interpretation and discussion of the results. NC is responsible for engineering management, maintenance, and
hardware and software upgrades of the Falkland Islands Radar. PB and TB are responsible for operational data management, data curation,
and quick-look data visualisation software for the Falkland Islands Radar. All the authors contributed to the final draft of the paper.

*Competing interests.* One of the co-authors (Andrew J Kavanagh) is a member of the editorial board of Annales Geophysicae.

*Acknowledgements.* The Falkland Islands SuperDARN radar (FIR) was originally funded by the UK Natural Environment Research Council
(NERC) through grants NE/G018707/1 and NE/G019665/1. It is now funded as part of the British Antarctic Survey (BAS) Polar Science for
Planet Earth programme and by NERC grant NE/R016038/1. We thank Mervyn Freeman and Steve Milan, who were the original FIR PIs
when these data were acquired. We thank members of the engineering team at La Trobe University, Melbourne, Australia, particularly John
Devlin, Darrell Elton, and Brian Bienvenu, for design of the instrument electronics and operating system. We thank BAS Engineering staff,

and particularly Octavian Carp, for maintenance and instrument upgrades. We thank staff at the BAS Stanley Office for administrative help regarding the radar management.

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
