# Peer review of "High-time resolution analysis of meridional tides in the upper mesosphere and lower thermosphere at mid-latitudes measured by the Falkland Islands SuperDARN radar"

_Annales Geophysicae, 2023_

## Author Response (AR1)

**Response to reviewers:**

**Angeo-2023-15**

**High-time resolution analysis of meridional tides in the upper mesosphere and lower thermosphere at mid-latitudes measured by the Falkland Islands SuperDARN radar**

**Gareth Chisham et al.**

Text in italics represents the reviewer's comments, with our response below. All line numbers given refer to the annotated version of the paper, where new text that has been added to the paper is in bold type.

**Response to Reviewer 1**

*The authors identified meteor echoes as those having altitude ranges of about 100 km and low spectral width, based on the results by Chisham and Freeman (2013). I have a concern about the possibility of the contamination of non-meteor echoes, such as E-region echoes, sporadic E echoes, and PMSEs. They have not fully discussed these possibilities, especially because they deal with echoes with Doppler velocities larger than 100 m/s. In addition to the echoes I already mentioned, there are other echoes such as HAIR (high aspect angle irregularity region, Milan et al., 2003) and FAIR (far aspect angle irregularity region, St.-Maurice and Nishitani, 2020), both located in the lower E-region ionosphere around 100 km altitude. They are not meteor echoes and do not always move with the ambient neutral wind. Some of them have similar characteristics as meter echoes (e.g., located around 100 km altitude and having low spectral width). Therefore, possible contamination of these echoes (E-region, sporadic E-region, PMSE, HAIR, and FAIR echoes) should be discussed in the text.*

To address the reviewer's concerns, we now include a major discussion in the instrumentation section on the potential contamination of the meteor echoes with other echo types (**lines 89-148**). This also includes a new figure (**Figure 1**) which highlights different echo populations seen by the FIR SuperDARN radar at near ranges, and how these can be distinguished from each other.

*The most promising way to solve the problem of distinguishing meteor echoes is to obtain the raw time series data, as reported by Yukimatu and Tsutsumi (2002) and Tsutsumi et al. (2009). Meter echoes should appear in the raw time series data as the echoes with a sudden increase of the echo power, followed by its exponential decay. Chisham and Freeman (2013) argue that the standard SuperDARN radars do not record raw time series data. However, the function of recording raw time series data has been implemented in the standard SuperDARN radar operation software, and several SuperDARN radars actually record raw time series data. I do not require the authors to analyze the raw time series data in the current study. However, it is not the right manner to ignore previous studies completely. Works related to raw time series data analysis should be introduced in the discussion.*

The reviewer is right that our paper should discuss the previous methods that have been used to measure SuperDARN meteor echoes at high-time resolution. We have added new text to the Introduction section (**lines 36-39**) to address this oversight.

*Line 173 "persistant" should be "persistent"*

This has been corrected (**line 230**).

**Response to Reviewer 2**

*In this paper, the authors presented high-resolution analysis of meridional winds obtained from meteor echoes observed by the superdarn radar. High-resolution (1 min) continuous wind observations are very important for addressing the influence of the entire spectrum of waves on the MLT region. However, the data procedure is vaguely described.*

We don't fully agree that the data procedure is vaguely described. The data procedure is standard as for many previous SuperDARN meteor scatter papers. Our paper includes a concise explanation of the meteor echo analysis and is well referenced regarding the measurements of meteor echoes with SuperDARN. Further detail is easily found in the referenced articles. However, we hope that the extended instrumentation section (**lines 89-148**) goes some way to addressing the reviewer's concerns.

*How many underdense meteors can be detected per minute or per hour? Need to include a plot showing time variation of meteor counts.*

This is now discussed explicitly in the paper (**lines 112-123**) and presented in the new figure (**Figure 1a**).

*The authors need to show clearly how meteor echos and E-region echoes are differentiated, with a few examples.*

We hope that the extended instrumentation section (**lines 89-148**) addresses the reviewer's concerns about the different echo types observed by the FIR SuperDARN radar at near ranges, and that the new figure (**Figure 1**) provides a demonstration of where different echo populations occur, and what the potential for contamination of meteor echoes at FIR is.

---

## Author Response (AR2)

**Response to reviewer**:

**Angeo-2023-15**

**High-time resolution analysis of meridional tides in the upper mesosphere and lower thermosphere at mid-latitudes measured by the Falkland Islands SuperDARN radar**

**Gareth Chisham et al.**

Text in italics represents the reviewer's comments, with our response below. All line numbers given refer to the annotated version of the paper, where new text that has been added to the paper is in bold type.

**General Response**

Due to the multiple potential scattering targets within the SuperDARN fields-of-view, with all SuperDARN studies there is always a problem with contamination of the signal with echoes from unwanted targets. This is particularly true with the contamination of ionospheric F-region scatter with ground and sea echoes. The same issue exists in near-range studies between meteor echoes and ionospheric E-region scatter.

Even given the large amount of effort that has been expended over the last few decades to filtering data to remove ground and sea scatter, the methods to separate the different types of scatter are imperfect, and despite these best efforts, almost always some contamination will remain. The overlap of different populations in near-range scatter has not been given the same attention and a well-proven method of successfully separating echo populations at near ranges is still elusive. This is something that would be of great benefit for the SuperDARN community and should be a topic of future study.

In the absence of such a well-proven and reliable method of echo separation, we feel that the questions that need to be asked in studies like this one are: (1) How much contamination is there and how does this typically vary with range? (2) How can we remove this contaminating backscatter to the best of our ability? and (3) What is the effect of any remaining contamination on the signals that we wish to study. The most significant of these regarding any scientific output is question (3), and this is something that we have presently failed to address in this paper. To rectify this we have added additional discussion of this aspect to the discussion section of the paper (**lines 292-316**). Our conclusion in this paper is that the effect of any non-meteor echo contamination on the tidal signals would be to reduce the amplitude of the signals. Hence, the fact that we still see strong tidal signals for much of the year suggests that this contamination is minimal at the times where we observe strong tidal signals.

**Specific Responses**

**Reviewer:** *The manuscript has been revised. Unfortunately, I still have a few concerns about its content. I will describe the points of my concerns below.*

*Lines 95-100: The echoes measured by FIR are different to those observed at auroral latitudes. FIR is located at mid-latitudes, and near range FIR echoes originate from ~55◦S geographic latitude (~40◦S geomagnetic latitude). This is a significant distance from the auroral region, which only appears in farther ranges of the FIR field-of-view during disturbed times. Hence, E-region echoes are rarer than at auroral latitudes, especially during the solar minimum interval studied here. Visual inspection of daily scatter plots from FIR indicates that*

*the near ranges are overwhelmingly dominated by meteor echoes. The main potential contaminants are sea scatter at farther ranges, and PMSE at the nearest ranges. E-region echoes are much rarer.*

*I have a serious concern about this paragraph in two aspects.*

*The authors say that the E-region echoes observed from ~ 40 S geomagnetic latitude are much rarer than those at auroral latitude – this statement is not valid. There are significant amounts of echoes at this mid-latitude range.*

**Response:** The reviewer makes a fair comment about our statement regarding the differences between the amount of E-region echoes seen at auroral latitudes and mid-latitudes. Our statement was based purely on our own experiences of looking at a very large number of daily time-series plots from the FIR radar, for which our observation "*Visual inspection of daily scatter plots from FIR indicates that the near ranges are overwhelmingly dominated by meteor echoes*" still stands (meteor echoes are very distinctive when compared to E-region echoes). However, our familiarity is with this single mid-latitude radar alone, and not with others. Hence, our statement regarding "*E-region echoes are rarer than at auroral latitudes*" is not backed up by any rigorous analysis, even though our experience is that this is the case with the FIR data compared to the auroral SuperDARN radars that we are familiar with. Consequently, we have removed most of the paragraph highlighted by the reviewer (**lines 92-103**) and totally rephrase our discussion of contaminating factors (**lines 147-160**). We also now discuss that the characteristics of E-region echoes at mid-latitudes may be different to those observed in the auroral zone (**lines 154-158**).

**Reviewer:** *See the paper by Yakymenko et al. (2015)., for example. They conducted a statistical analysis of the occurrence of E-region echoes at the Hokkaido East radar (HOK), which is located at a similar geomagnetic latitude (with opposite sign) as FIR. Yakymenko et al. (2015) show that HOK observes E-region echoes with a maximum occurrence rate of 0.3 to 0.4 (see Figures 3,4 and 5). These rates cannot be regarded as "much rarer than at auroral latitudes."*

**Response:** Figure 1 of Yakymenko et al. (2015) does clearly show a clear band of E-region scatter for much of the day, hence we do not doubt the existence of extended intervals of E-region scatter in the HOK data set. However, we rarely see sustained patches of E-region scatter like this in the FIR data set. We are presently uncertain of the reasons for these differences. However, we would question whether the statistics presented in the Yakymenko paper include significant contamination from meteor echoes which might have impacted the maximum occurrence rates presented. The study only explicitly excludes potential meteor echo data from ranges 0 and 1, but it is well known that meteors are ubiquitous in SuperDARN ranges up to ~600km. There is no obvious attempt to remove these echoes at other ranges, which would be a contaminating factor in a study of E-region scatter.

There are four reasons that suggest to us that the Yakymenko study may include significant meteor echo contamination: (1) Meteor echo occurrence would be expected not to vary much with changing beam, in contrast to E-region echoes that vary along a constant aspect angle. The fact that the occurrence distributions presented by Yakymenko et al. (2015) are relatively constant with beam direction (their figures 3 and 4, and their comment "*Surprisingly for some months, the band of enhanced short-range echo occurrence is almost a straight stripe*"), and moves across the aspect angle height curves with increasing beam, suggest that their statistics are potentially contaminated by meteor scatter. (2) The occurrence variations with local time and time of year (their figure 5) match very closely to the variations

seen for meteor echoes at FIR by Hibbins et al. (2011). (3) The lack of correlation with geomagnetic indices (their figure 6) is also more representative of meteor echoes, for which there is little or no dependence on geomagnetic activity. (4) The clear observation of a semi-diurnal tide in the velocity statistics (their figure 9) suggests that the neutral wind variations are dominant in the statistics. The most likely explanation for this is a high-level of contamination by meteor echoes.

Consequently, our argument is that any contamination by meteor echoes in the work of Yakymenko et al. (2015) would inflate the occurrence rate estimates presented there. Hence, we would question whether the quoted values are a reliable indicator of the typical occurrence rates for E-region scatter at mid-latitudes. As a consequence, we stand by our statement that mid-latitude "*E-region echoes are rarer than at auroral latitudes*" which predominantly results from our experience of studying large numbers of daily scatter variations at FIR. However, as stated above, we cannot presently back up this inference with any rigorous analysis, hence we have removed this statement from the text of the paper. This ambiguity between E-region and meteor scatter at near ranges is something that requires significant further study, both at mid and high latitudes, and we now discuss this shortcoming of studies of near-range scatter in the discussion section of the paper (**lines 292-316**).

**Reviewer:** *In addition, at mid-latitudes, the magnetic field lines are more inclined to the horizontal direction, locating the preferable location of E-region echoes closer to the radar than the high-latitude radar echoes. Consequently, the E-region echo regions are more overlapped with the meteor echo regions.*

*From Figure 1, it is difficult to see how much overlapping exists between A and C regions. If the authors want to say that the situation differs from the HOK case, they should clarify how and why they differ.*

**Response:** We agree that it is difficult to see how much overlapping occurs between regions A and C in figure 1, and now discuss this in the text (**lines 139-160**). At the present time, we do not know if, and why, there are differences between the FIR and HOK meteor and E-region observations. However, a full comparative analysis of near-range scatter at different mid-latitude SuperDARN stations is beyond the scope of this present work.

**Reviewer:** *One possible way to distinguish between the E-region and meteor echoes is to see the aspect angle, such as Liu et al. (2013), which examined the aspect angle condition to distinguish between E-region echoes and PMSEs. If the authors cannot use the method by Yukimatu et al. (2002) and Tsutsumi and Yukimatu (2009), this technique might be helpful.*

**Response:** As the boresight direction for the FIR radar is perpendicular to lines of constant geomagnetic latitude, there is very little variation in the aspect angle condition across the different beams in the field of view. Hence, it would be very difficult to use this to distinguish from variations occurring at a constant range.

**Reviewer:** *The authors say that the echoes measured by FIR differ from those observed at auroral latitudes. I am afraid that this statement will raise a serious concern about the validity of applying previous algorithms by Chisham and Freeman (2013) and Chisham (2018) because they use only high-latitude SuperDARN (Saskatoon) data to distinguish between meteor echoes from other echoes, such as E-region backscatter echoes.*

**Response:** Our statement "*The echoes measured by FIR are different to those observed at auroral latitudes*" has been misunderstood by the reviewer. Our meaning was that different

types of echoes are measured at mid-latitudes than at higher latitudes, most pertinently a reduction in the amount of F-region ionospheric scatter observed. Our experience working with FIR has also suggested to us a reduction in the amount of E-region scatter at mid-latitudes compared to that seen by higher-latitude radars. However, in the absence of relevant statistics we have decided to remove this statement from the text.

**Reviewer:** *Mid-latitude E-region echoes have lower velocities and eventually lower velocity error values than high-latitude E-region echoes. Then, it might be more difficult to distinguish between them.*

*The author should describe how the Chisham and Freeman (2013) algorithm can be applied to the mid-latitude radar echoes, which have different characteristics from the high-latitude radar echoes.*

**Response:** We still view the previous algorithm of Chisham and Freeman (2013) as a valid pre-processing tool for mid-latitude scatter, although we appreciate that if the character of E-region scatter at mid-latitudes is significantly different to that seen at higher latitudes, as suggested by the work of Yakymenko et al. (2015), then the method may not be as efficient at reducing the E-region contamination in the FIR near-range scatter. We now discuss this in the text (**lines 148-160**).

**Reviewer:** *Figure 1: Please describe the relation between the local time and UT.*

**Response:** As the FIR radar is located at a geographic longitude of 59.0ºW, the difference between universal time and local time is ~4 hours. We now clearly state this in the paper (**lines 78-79**).